# Boosting the Capacitive Performance of Supercapacitors by Hybridizing N, P-Codoped Carbon Polycrystalline with Mn_3_O_4_-Based Flexible Electrodes

**DOI:** 10.3390/nano13142060

**Published:** 2023-07-12

**Authors:** Yu-Min Kang, Wein-Duo Yang

**Affiliations:** Department of Chemical and Materials Engineering, National Kaohsiung University of Science and Technology, Sanmin District, Kaohsiung City 807, Taiwan; f110146142@nkust.edu.tw

**Keywords:** chitosan, carbon skeleton, Mn_3_O_4_, N, P-codoped carbon polycrystalline, supercapacitor

## Abstract

Chitosan, a biomass raw material, was utilized as a carbon skeleton source and served as a nitrogen (N) atom dopant in this study. By co-doping phosphorus (P) atoms from H_3_PO_4_ and nitrogen (N) atoms with a carbon (C) skeleton and hybridizing them with Mn_3_O_4_ on a carbon fiber cloth (CC), an Mn_3_O_4_@NPC/CC electrode was fabricated, which exhibited an excellent capacitive performance. The N, P-codoped carbon polycrystalline material was hybridized with Mn_3_O_4_ during the chitosan carbonization process. This carbon polycrystalline structure exhibited an enhanced conductivity and increased mesopore content, thereby optimizing the micropore/mesopore ratio in the electrode material. This optimization contributed to the improved storage, transmission, and diffusion of electrolyte ions within the Mn_3_O_4_@NPC electrode. The electrochemical behavior was evaluated via cyclic voltammetry and galvanostatic charge–discharge tests using a 1 M Na_2_SO_4_ electrolyte. The capacitance significantly increased to 256.8 F g^−1^ at 1 A g^−1^, and the capacitance retention rate reached 97.3% after 5000 charge/discharge cycles, owing to the higher concentration of the P-dopant in the Mn_3_O_4_@NPC/CC electrode. These findings highlight the tremendous potential of flexible supercapacitor electrodes in various applications.

## 1. Introduction

Supercapacitors (SCs) have emerged as promising energy storage devices due to their durability, rapid charge and discharge rates, high power density, good reversibility, low maintenance cost, and environmental friendliness [1,2]. Typically, carbon materials are used as the electric double-layer capacitor material to provide power, while pseudo-capacitance materials are employed to enhance the energy storage capacity and expand the potential window of capacitors [3,4,5].

Among the various supercapacitor electrode materials, MnO_x_ is particularly attractive [6]. Manganese ions, including Mn^2+^, Mn^3+^, and Mn^4+^, are present in different manganese oxides, such as MnO, MnO_2_, Mn_2_O_3_, and Mn_3_O_4_ compounds [7]. Specifically, Mn_3_O_4_ (Mn^2+^(Mn^3+^)_2_O_4_) adopts a spinel structure (AB_2_O_4_) and encompasses multiple oxidation states of Mn^2+^ and Mn^3+^ [8]. Furthermore, Mn_3_O_4_ exhibits a high theoretical capacitance and the lowest oxygen evolution reaction activity among the MnO_x_ materials [9]. In a recent study, Sahoo et al. [10] demonstrated the utilization of rGO@Mn_3_O_4_ electrodes, which enabled the exploration of multivalent redox states (+2, +3, and +4) of Mn and achieved a wide operating potential of 1.2 V (vs. Ag/AgCl). These excellent properties, including a wide potential window and pseudo-capacitive behavior, make Mn_3_O_4_ highly promising for the development of flexible aqueous asymmetric supercapacitors (FAASs).

Carbon materials, such as porous carbon, graphene, and carbon nanotubes, have limitations in terms of their specific capacitance [11,12,13], while manganese oxides suffer from a low conductivity [14]. Moreover, the development of wearable and flexible electronic devices has emphasized the need for flexible and portable energy storage devices [15]. To overcome these challenges, hybrid nanostructured electrodes combining manganese oxides and carbon materials have been extensively researched. In such hybrid systems, carbon materials not only contribute to the electric double-layer capacitance, but also enhance the conductivity of manganese oxides [16].

To improve the electrochemical performance, various approaches have been explored, including the combination of Mn_3_O_4_ with conductive substrates such as activated carbon [17]. Furthermore, the combination of graphene nanosheets with Mn_3_O_4_ has been investigated as a hybrid electrode [18]. Jiang et al. [19] successfully synthesized self-assembled Mn_3_O_4_/carbon cloth (Mn_3_O_4_/CC) electrodes using a two-step hydrothermal method. These Mn_3_O_4_/CC electrodes were tested in a three-electrode system using Na_2_SO_4_ and KOH as electrolytes. It was observed that Mn_3_O_4_/CC exhibited a higher specific capacitance and lower internal resistance in the KOH electrolyte. In order to achieve high-performance electrodes for supercapacitors, carbon materials are commonly hybridized with high-capacity metal oxides as substrates [20]. It is recognized that the introduction of heteroatoms (such as N, S, O, B, and P) into carbon-based materials has a significant impact on the conductivity, wettability, and pseudo-capacitance of electrode materials [21]. Moreover, the synergistic effect of multi-heteroatom-doped carbon materials on electrochemical performance has been found to be superior to that of single heteroatoms [22].

Various zero-, one-, or two-dimensional heteroatom-doped carbon materials and metal oxide composites have been reported [23]. However, these materials often suffer from weak mechanical properties, a low specific surface area, and a low porosity, which hinder the penetration of electrolytes and the diffusion of ions into the internal structure of electrode materials. To address these challenges, recent research has highlighted the use of heteroatom-doped porous carbon materials with a 3D interconnected network structure. Such a 3D network possesses a highly open structure and abundant pores, facilitating rapid electron transfer and electrolyte ion penetration into electrode materials, thereby enhancing the overall reaction kinetics. Additionally, a 3D structure can effectively absorb external strain and reduce resistivity during mechanical deformation, making it suitable for flexible energy storage devices [24]. However, the simple combination of carbon and Mn_3_O_4_ often leads to the aggregation of Mn_3_O_4_ nanoparticles, resulting in sluggish electron and ion migration kinetics [17]. Therefore, it is crucial to optimize the microstructure and morphology of a composite to improve the Faradaic response in the surface and near-surface regions of an electrode [25].

Chitosan, as a highly abundant biomass material, possesses an internal, interconnected network nanostructure that holds great potential for practical applications. Chitosan contains -NH_2_, -OH, and a small amount of acetamide groups, as shown in Appendix A. The carbonyl group (-COOH) contained in acetic acid can combine with -NH_2_ to promote the solubility of chitosan. At the same time, in acidic conditions, chitosan has a positive ion charge of -NH_3_^+^, which can be chemically combined with anion-bearing metal ions, such as PO_4_^3−^, for chemical modification [26]. In addition, phosphoric acid is an activator in the carbonization process of chitosan, which greatly increases the specific surface area of the resulting carbon material. However, phosphoric acid is a strong acid and can only be a dopant source of P in small amounts. Excessive amounts will easily cause the dissolution of the chitosan and Mn(OH)_2_.

Wei et al. [27] reported that a sandwich-like chitosan porous carbon sphere/MXene composite possesses a high specific capacitance and rate performance as a supercapacitor. Ding et al. [28] conducted N and P co-doping into the carbon framework of chitosan to enhance its electrical conductivity and improve the compatibility of carbon materials with Mn_3_O_4_ and electrolytes. The fabricated Mn_3_O_4_ nanosheets provided abundant active regions, while the C-O-Mn bonds formed with N, P-codoped carbon (NPC) facilitated fast electron transfer kinetics. The intercalation pseudocapacitive mechanism involving Mn^3+^/Mn^4+^ conversion in Mn_3_O_4_@NPC not only extended the potential window to 0–1.3 V (vs. Ag/AgCl), but also significantly enhanced the pseudo-capacitance. Sun et al. [20] employed manganese nitrate (Mn(NO_3_)_2_) as a raw material to prepare a 3D N, P-codoped hierarchical porous carbon framework using a one-step method. They embedded Mn_3_O_4_ nanoparticles in situ to obtain an Mn_3_O_4_@NPC material as a supercapacitor electrode, which exhibited an excellent capacitance performance and cycle stability. Additionally, they reported a direct heat treatment method utilizing inexpensive melamine sponge, manganese nitrate (Mn(NO_3_)_2_), and biomass salt. The resulting Mn_3_O_4_-embedded N, P-codoped carbon sheets demonstrated an outstanding electrochemical performance [21].

Graphene and carbon nanotubes have been found to enhance the dispersion of Mn_3_O_4_ powder effectively. However, graphene materials are expensive and the preparation of doped materials requires additional dopant sources, thereby increasing the complexity and cost. On the other hand, chitosan, a cost-effective material, contains heterogeneous impurity elements such as N. Therefore, utilizing chitosan as a framework for Mn_3_O_4_ carbon composite electrodes has garnered attention among researchers. It is worth noting that the research in this area is relatively limited compared to graphene-based studies. Thus, investigating the use of chitosan as a skeleton for Mn_3_O_4_ carbon composites holds significant potential and is an interesting research topic. The novelty of this study is its investigation, in detail, of the effect of the doping amount of P on the crystalline phase, microstructure, and powder properties of a prepared N, P-codoped carbon hybridized with Mn_3_O_4_ (Mn_3_O_4_@NPC) electrode material. As a result, the electrochemical performance of the electrode was greatly improved by the amount of P-dopant.

In this study, carbon fiber cloth (CC) was used as the substrate, H_3_PO_4_ was used as the P doping source, chitosan was used as the N doping source, and Mn_3_O_4_ was hybridized on the chitosan-carbonized hierarchical porous carbon skeleton to prepare the Mn_3_O_4_@NPC electrode material. Due to the bendability of CC, the prepared Mn_3_O_4_@NPC/CC electrode could be applied to flexible supercapacitors.

## 2. Materials and Methods

### 2.1. Synthesis of Mn_3_O_4_@NPC/CC Hybrid Electrode

First, carbon fiber cloth (CC) was cut into a size of 2 cm × 2 cm to serve as the substrate. It was then soaked in deionized (DI) water and cleaned for 10 min using an ultrasonic cleaner. Subsequently, the CC was taken out and soaked in ethanol, followed by another 10 min of cleaning using an ultrasonic oscillator. This cleaning process was repeated twice to ensure the removal of debris and dirt from the CC. The cleaned CC was placed in a tubular furnace and heated at 10 °C min^−1^ to 400 °C for 3 h to increase its roughness and introduce oxygen-containing functional groups onto the CC surface. Afterward, the heated CC was cleaned in alcohol again using an ultrasonic oscillator for 10 min. Finally, the CC was taken out and placed in a vacuum drying oven.

For the synthesis of the Mn_3_O_4_@NPC/CC hybrid electrode, 1.0 g of chitosan was dissolved in 20 mL of DI water. A small amount of acetic acid was added to promote dissolution, and then 20 mL of 0.1 M manganese nitrate solution was slowly added to the chitosan solution, with stirring for 1 h to produce a complex of chitosan and Mn(OH)_2_ (precursor of Mn_3_O_4_). The CC substrate was immersed in the solution to ensure the direct growth of Mn_3_O_4_ on its surface. Next, 20 mL of a 1.0 M NaOH solution was gradually added to the solution in batches, and stirring was continued for 1 h. Subsequently, 1.0 M dilute H_3_PO_4_ (2 mL, 5 mL, or 10 mL) was added as a source of P-dopant, and the solution was stirred at room temperature for 2 h. Three different amounts of H_3_PO_4_ were used to prepare the N, P-codoped carbon materials, which were designated as NPC(L), NPC(M), and NPC(H), respectively.

The stirred solution was transferred to a Teflon-lined stainless-steel autoclave and subjected to a hydrothermal process at 180 °C for 10 h. After the hydrothermal treatment, the sample was freeze dried for 24 h to obtain Mn_3_O_4_@NPC/CC. The dried Mn_3_O_4_@NPC/CC was then annealed in a tube furnace at 10 °C min^−1^ to 600 °C for 1 h under a N_2_ atmosphere, followed by natural cooling to room temperature. The mass of Mn_3_O_4_ in each sample was 0.01 (±0.001) g; if the mass difference was too large, it would affect the specific capacitance. Therefore, this study set the quantity within a fixed range to restrain its error. Figure 1. illustrates the fabrication process of the Mn_3_O_4_@NPC/CC hybrid electrode.

In this study, 15 mL, 20 mL, and 25 mL of H_3_PO_4_ were added to the colloidal precipitate containing Mn(OH)_2_ and chitosan and stirred for 2 h. It was obvious that the dissolution of the mixtures occurred, and the volume of the colloidal precipitate reduction was approximately 15–20 vol.%. Thus, the mass of the active material (Mn_3_O_4_) was greatly reduced after adding 15 mL of H_3_PO_4_ for the fabrication of Mn_3_O_4_@NPC. Moreover, its specific capacitance value was also seriously attenuated (Appendix A). Therefore, the addition of 10 mL of H_3_PO_4_ was the maximum amount.

### 2.2. Characterization

Several characterization techniques were employed to examine the properties and microstructures of the Mn_3_O_4_-based materials.

Field-emission scanning electron microscopy (FESEM), using a JEOL6330 (Tokyo, Japan) instrument with an acceleration voltage of 80 kV, was utilized to examine the microstructures of the as-prepared Mn_3_O_4_-based materials. This technique provided a detailed analysis of the surface morphology of the materials. Additionally, for a higher magnification and more detailed microstructural analysis, high-resolution transmission electron microscopy (HRTEM), using a JEOL TEM-3010 (Tokyo, Japan) instrument with an acceleration voltage of 80 kV, was employed. The crystal phases present in the materials were identified using an X-ray diffraction (XRD) analysis. A Bruker D8 ADVANCE (Karlsruhe, Germany) instrument, equipped with Cu as the anode (Cu K_α_ radiation, λ = 1.5406 Å) and tungsten (W) filament as the cathode, was used for the XRD measurements. The Shirley baseline was applied to fit the baseline of the diffraction spectrum. This technique allowed for a determination of the crystal structure and phase composition of the materials.

X-ray photoelectron spectroscopy (XPS) was conducted to determine the surface composition and binding energy of the Mn_3_O_4_-based materials. A PHI 5000 VersaProbe (Tokyo, Japan) instrument was used for the XPS measurements. The binding energy of C at 284.6 eV was used as a reference for calibrating the charge-shifted energy scale. Gaussian peak deconvolution was performed to identify the chemical components in the spectra.

The specific surface area of the Mn_3_O_4_-based materials was determined using the Brunauer–Emmett–Teller (BET) method. The mesopore area was analyzed using the Barrett–Joyner–Halenda (BJH) method. Additionally, the t-plot method was employed to determine the micropore area. N_2_ adsorption–desorption isotherms were obtained using a Micrometrics ASAP 2020 instrument (Micrometrics, Atlanta, GA, USA) for the characterization of the as-obtained Mn_3_O_4_-based materials. The contact angle of the Na_2_SO_4_ aqueous electrolyte on the electrode surface was measured using a contact angle goniometer (NBSI OSA60, Ningbo, China). The powder (Mn_3_O_4_ or Mn_3_O_4_@NPC) was formulated into a uniform dispersion according to a certain proportion (5 wt.%), and then the dispersion was dropped on a watch glass and placed in a vacuum oven. After the water was removed, the measured liquid (Na_2_SO_4_ electrolyte) was dropped on the powder, and the contact angle between the powder and electrolyte was measured. This provided information about the wettability and surface properties of the as-prepared electrodes.

These characterization techniques provided valuable insights into the microstructures, crystal phases, surface properties, and pore characteristics of the Mn_3_O_4_-based materials, contributing to a comprehensive understanding of their properties and performance.

### 2.3. Electrochemical Properties of Mn_3_O_4_@NPC/CC Hybrid Electrode

All the electrochemical characteristics were determined using a three-electrode electrochemical cell setup. The Mn_3_O_4_/CC hybrid electrode (2.0 cm × 2.0 cm) fabricated in this study served as the working electrode, while an Ag/AgCl electrode was used as the reference electrode. A platinum plate (2.0 cm × 2.0 cm) was employed as the counter electrode. The electrolyte used was a 1.0 M Na_2_SO_4_ aqueous solution. Cyclic voltammetry (CV) and galvanostatic charge–discharge (GCD) tests were conducted using a CHI 760D electrochemical workstation.

The specific capacitance (C_m_) was calculated based on the discharge curve obtained from the GCD test, using Equation (1) [29]:(1)$$C_{m}=\frac{i\times \Delta t}{\Delta V\times m}\;$$

Here, i (A) represents the discharge current, Δt (s) denotes the discharge time, ΔV (V) represents the potential difference during the discharge, and m (g) is the mass of the loaded active material (Mn_3_O_4_).

## 3. Results

### 3.1. The Physical Properties of N, P-Doped Mn_3_O_4_ Material

Figure 1 shows the SEM analysis of the Mn_3_O_4_-based powders prepared under various conditions. The SEM image of Mn_3_O_4_ reveals irregularly shaped particles that were stacked on crystal grains and formed a porous nanostructure (Figure 1a). When chitosan was added as the carbon precursor to prepare Mn_3_O_4_@C, the SEM observations show the presence of flaky and irregular particles in Mn_3_O_4_@C (Figure 1b). The distinct shapes and structures observed in the prepared Mn_3_O_4_ and carbon materials will be further analyzed using HRTEM in subsequent studies.

The morphology of the prepared Mn_3_O_4_ was closely related to the amount of H_3_PO_4_ added during the preparation, as shown in Figure 1c–e. The SEM image of the Mn_3_O_4_ prepared with a lower amount of H_3_PO_4_ reveals a flower-like arrangement and a sheet-like structure. In the SEM image of the Mn_3_O_4_ obtained with a moderate amount of H_3_PO_4_ (Figure 1d), the sheet structure exhibits curly edges, along with some strip-shaped crystal structures. Figure 1e displays the image of the Mn_3_O_4_ prepared with a higher amount of H_3_PO_4_. The SEM analysis reveals an increased thickness in the flakes with curled edges and the formation of strip-like structures. Figure 1f–h depict the SEM analysis of the Mn_3_O_4_@NPC/CC electrode prepared by adding different amounts of H_3_PO_4_. The SEM image of the Mn_3_O_4_ obtained with a lower amount of H_3_PO_4_ (Mn_3_O_4_@NPC(L)/CC) electrodes shows regularly arranged flower-like and sheet-like structures on carbon fibers (Figure 1f). The fabricated Mn_3_O_4_@NPC(M)/CC electrode demonstrates sheet-like and strip-like structures with curled edges, which are evenly adhered to the carbon fiber cloth (Figure 1g). Similarly, the Mn_3_O_4_@NPC(H)/CC electrode exhibits sheet-like and strip-like structures of Mn_3_O_4_ attached to the carbon fiber cloth, which are uniformly distributed (as shown in Figure 1h).

Figure 2 shows the XRD patterns and contact angles (CA) of the various Mn_3_O_4_-based materials. From the XRD diffraction peaks of the carbonized chitosan, broad and low diffraction peaks were observed at 23.65° and 44.04°, corresponding to the (002) and (100) crystal planes of the carbon. This confirms the successful conversion of chitosan into an amorphous (very low crystalline) carbon material. In contrast, the XRD exhibits sharp and narrow diffraction peaks with few impurity phases of Mn_3_O_4_, revealing that the prepared Mn_3_O_4_ powder had a good crystallinity, but the powder particles were relatively coarse (consistent with the SEM results in Figure 1). The diffraction peaks were observed at 18.13°, 29.05°, 31.2°, 32.5°, 36.24°, 38.19°, 44.53°, 50.91°, 58.67°, 60.01°, and 64.8°, corresponding to the (101), (112), (200), (103), (211), (004), (220), (204), (321), (224), and (400) crystal planes of the Mn_3_O_4_, respectively (according to the JCPDS card No. 24-0734) [30].

However, after chitosan was added to prepare the Mn_3_O_4_@C material and underwent carbonization, the diffraction peaks of the Mn_3_O_4_@C hybrid material became boarder and shifted a little to a lower 2θ, indicating that the Mn_3_O_4_@C electrode material had smaller particle sizes. There were no distinct diffraction peaks corresponding to the carbon crystals, indicating that carbon also existed in an almost amorphous form in the Mn_3_O_4_@C hybrid material. The Mn_3_O_4_@NPC materials also exhibited weaker and broadened diffraction peaks of the Mn_3_O_4_. However, in a distinct difference, the diffraction peaks of the carbon appeared at (002) and (100), indicating that a N, P-codoped carbon (NPC) polycrystal was established by the introduction of phosphorus dopants. With an increase in the amount of P doping, the peaks of the N, P-codoped carbon became stronger and narrower. On the other hand, the diffraction peaks of the Mn_3_O_4_ (on Mn_3_O_4_@NPC) were broadened and shifted a little to a lower 2θ, similar to the XRD of Mn_3_O_4_@C (Figure 2a). The XRD diffraction peak of the Mn_3_O_4_ material was broadened and slightly shifted, showing that the particle size of the powder was smaller, which is also consistent with the SEM analysis in Figure 1.

Based on the above XRD and SEM analyses, it is speculated that the mixture of chitosan and Mn(OH)_2_ (the precursor of Mn_3_O_4_) was based on the N atom in the amino group (-NH_2_) of chitosan chelating with the manganese ions in Mn(OH)_2_. This enabled the heat-treated Mn_3_O_4_ to be fully dispersed in amorphous carbon (derived from chitosan carbonization) and produced smaller sizes of the Mn_3_O_4_@C material. In addition, phosphoric acid was not the only source of P-dopant; phosphoric acid also played the role of activator in the process of chitosan carbonization, thus making the prepared carbon more active in the carbonization, greatly increasing the specific surface area of the NPC and promoting the transformation of the NPC from the original amorphous carbon into a polycrystalline structure. Moreover, the H_3_PO_4_ dissolved Mn_3_O_4_ into smaller particles. As a result, it can be anticipated that Mn_3_O_4_@NPC would possess a larger specific surface area.

The contact angles of the Na_2_SO_4_ electrolyte on Mn_3_O_4_/CC and Mn_3_O_4_@NPC(H)/CC were 40.1° and 20.6°, respectively (Figure 2b). This discrepancy can be attributed to the presence of Mn_3_O_4_@NPC(H)/CC, where the carbon skeleton offered numerous N active sites and co-doped P atoms, leading to a synergistic effect. Additionally, the inclusion of P-O bonds within the carbon skeleton remarkably enhanced the wettability between the material and electrolyte.

The structure of Mn_3_O_4_@NPC(H)/CC was further examined using TEM and the results are illustrated in Figure 3. Figure 3a shows the interlaced structure of the flakes and strips in Mn_3_O_4_@NPC(H)/CC, which resulted from the higher amount of the P-dopant. Compared to the SEM and XRD studies, the sheet-like structure of the material appears relatively loose, potentially due to the N/P atom doping. The electron diffraction images exhibit blurry diffraction rings, indicating that the prepared Mn_3_O_4_@NPC(H) possessed a poor crystallinity (shown in the bottom right corner of Figure 3a). This finding aligns with the XRD analysis results. Additionally, the HRTEM image reveals that a few of the outer layers of Mn_3_O_4_ were covered by the NPC phase, as depicted in Figure 3b. Furthermore, the lattice spacing region in the HRTEM image (Figure 3b) was subjected to fast Fourier transform (FFT), resulting in the diffraction pattern displayed in Figure 3c. The diffraction pattern of Mn_3_O_4_@NPC(H) exhibits ring-like patterns that closely resemble the diffraction diagram shown in Figure 3a. This confirms that the Mn_3_O_4_@NPC(H) possessed a lower relative crystallinity compared to the Mn_3_O_4_@C.

The presence of heteroatoms embedded in the carbon skeleton to form NPC polycrystals affected the growth morphology and arrangement of Mn_3_O_4_. The incorporation of the NPC phase enhanced the structural stability of the electrode, shortened the transfer path, and increased the contact area between the electrode and electrolyte [31,32]. Additionally, it introduced new electroactive sites into the hybrid material, leading to an improved electrochemical performance of the Mn_3_O_4_@C/CC electrode. The inverse fast Fourier transform (IFFT) power spectrum was obtained from the lattice spacing region in the HRTEM image (Figure 3b). The response peaks in the spectrum correspond to the characteristic lengths of each lattice spacing in Figure 3d,e. By measuring the lattice spacing of the rod-shaped Mn_3_O_4_, it was determined to be 0.24 nm and 0.276 nm, respectively. This finding further confirms that the crystal phases correspond to the XRD patterns described in the previous section (Figure 2a). The diffraction peaks of this crystal aligned with Mn_3_O_4_@NPC(H), with characteristic peaks appearing at approximately 2θ = 36.41° and 32.55°. Based on Bragg’s Law, the interlayer spacing of the crystal planes was calculated to be 0.276 nm and 0.248 nm, respectively. Furthermore, according to the Miller index equation for the tetragonal crystal phase [33], $\frac{1}{d_{\left(hkl\right)}}=\frac{h^{2}+k^{2}}{a^{2}}+\frac{l^{2}}{c^{2}}$, the diffraction peaks corresponded to the (211) and (103) planes of Mn_3_O_4_. Simultaneously, the IFFT also clearly reveals that part of the Mn_3_O_4_ was covered by the NPC sheet. These results are consistent with the TEM analysis mentioned above. The observations indicate that Mn_3_O_4_@NPC(H), prepared with a higher amount of H_3_PO_4_, exhibited a poor crystallinity with broad and weak peaks. It can be concluded that Mn_3_O_4_@NPC(H) contains nanocrystals with less complete crystallization.

In Figure 4, the XPS analysis of the Mn_3_O_4_@NPC(H) material is presented. The full-range survey of XPS reveals the presence of five elements on the surface of the sample: Mn (10.8 at.%), O (25.4 at.%), C (61.4 at.%), N (2.3 at.%), and P (0.2 at.%). The characteristic peaks of the Mn elements indicate that the Mn 3s and Mn 3p peaks were satellite peaks of Mn 2p (Figure 4a). Two prominent binding energy peaks were observed at 654.06 eV and 642.26 eV, corresponding to the Mn 2p_1/2_ and Mn 2p_3/2_ peaks, respectively [34]. These peaks suggest the presence of Mn^2+^ and Mn^3+^ species (Figure 4b).

Figure 4c displays the binding energy spectrum of O 1s on the surface of the Mn_3_O_4_@NPC(H) material. The sharp peaks were analyzed using Gaussian deconvolution and divided into three peaks: 529.6 eV, 531.2 eV, and 533.0 eV. These peaks correspond to the binding energies of Mn-O-C/Mn-O-Mn [35,36,37], -OH of Mn-O-H [38], -C-O [39], and H-O-H [40], respectively. The clear Mn-O-C bond indicates the presence of a stable covalent bond between Mn_3_O_4_ and the carbon skeleton.

The results of the XPS region C 1s are shown in Figure 4d, which exhibits binding energy centered at 284.5 eV, 285.4 eV, and 288.7 eV, attributed to the C-C/C=C, C-N/C=O, and O-C=O bonding, respectively [41]. Among them, the stable bond between C and N can be known from the high bonding ratio of the C-N bond. In summary, the XPS analysis of Mn_3_O_4_@NPC(H) reveals the presence of nitrogen-containing functional groups from chitosan, providing a N source for the synthesis of Mn_3_O_4_, and demonstrates the hybridization of the P atoms with the carbon skeleton, indicating the successful incorporation of P into the Mn_3_O_4_@NPC electrode material.

To further investigate the powder properties of the porous, hierarchical, nanostructured Mn_3_O_4_@NPC, the BET specific surface area and pore size distribution of the fabricated Mn_3_O_4_ materials were analyzed. The N_2_ absorption/desorption isotherms of all the samples exhibited a typical type IV curve, indicating the presence of both mesopores and micropores in the materials’ structures (Figure 5a) [42]. The adsorption isotherm curve showed a steep rise at high relative pressures, and at a relative pressure of approximately 0.8~1.0, an H3-type desorption hysteresis loop was observed, indicating the presence of irregularities in the pore structure of the prepared Mn_3_O_4_@NPC material [43]. The incomplete closure of the curve’s tail can be attributed to the addition of chitosan as the carbon-based precursor. The Mn_3_O_4_@NPC materials prepared with different amounts of H_3_PO_4_ as the P-dopant source exhibited specific surface areas of 49.83 m^2^ g^−1^, 48.18 m^2^ g^−1^, and 86.16 m^2^ g^−1^, with corresponding pore sizes of 17.95 nm, 14.93 nm, and 8.31 nm, respectively (Table 1). Furthermore, an analysis of the pore size distribution reveals that the pore volume of the small-diameter pores in the Mn_3_O_4_@NPC increased with additional amounts of H_3_PO_4_ as the P-dopant source (Figure 5b). In fact, the introduced H_3_PO_4_ not only acted as the P-dopant source, but also acted as an activating agent, thereby increasing the porosity of the as-obtained materials.

Among the Mn_3_O_4_@NPC/CC electrode materials prepared with different amounts of H_3_PO_4_ as the P-dopant source, the Mn_3_O_4_@NPC(H)/CC electrode material exhibited the highest micropore area and pore volume. Additionally, it had the highest proportion of mesopore properties, thus enabling faster ion transport through the material.

Among the Mn_3_O_4_@NPC/CC electrode materials prepared with different amounts of H_3_PO_4_ as the P-dopant source, the Mn_3_O_4_@NPC(H)/CC electrode material exhibited the highest micropore area and pore volume. Additionally, it had the highest proportion of mesopore properties, thus enabling faster ion transport through the material. Moreover, the Mn_3_O_4_@NPC(H)/CC electrode material also possessed the highest specific surface area, which enhanced its capacitive performance [44].

### 3.2. Electrochemical Properties of Mn_3_O_4_@NPC/CC Electrodes

In the electrochemical characterization, a 1 M Na_2_SO_4_ aqueous solution served as the electrolyte, an Ag/AgCl electrode functioned as the reference electrode, and a Pt sheet was used as the counter electrode. The impact of varying the amounts of the P-dopant in the Mn_3_O_4_ electrode materials on their electrochemical properties was examined in this study. Figure 6 illustrates the CV curves obtained from the Mn_3_O_4_-based/CC electrodes. Figure 6a depicts the CV curve of the Mn_3_O_4_/CC electrode. The curve exhibits an asymmetric profile, indicating the electrode’s instability. Furthermore, as the scan rate increased, this instability became more pronounced, emphasizing the need for further investigation and optimization. As depicted in Figure 6b–d, two pairs of redox peaks were observed in the CV curves of the Mn_3_O_4_@NPC/CC electrodes at low scan rates. These peaks corresponded to the redox reactions of Mn^2+^/Mn^3+^ and Mn^3+^/Mn^4+^ of the Mn_3_O_4_@NPC/CC electrodes, respectively [45,46]. Even at higher scan rates, the CV curves maintained a non-rectangular shape with distinct redox peaks, indicating pseudocapacitive behavior [47]. Comparing the Mn_3_O_4_/CC electrode (Figure 6a) with the Mn_3_O_4_@NPC/CC electrode, it is evident that the area under the CV curve at a low scan rate was significantly enhanced. This confirms that the incorporation of the NPC polycrystalline phase into the Mn_3_O_4_-based electrode effectively enhanced the capacitance performance of the electrode.

Figure 7 shows the GCD curves of the various Mn_3_O_4_-based/CC electrodes at a current density of 1 A g^−1^. From the charge–discharge curve of the Mn_3_O_4_/CC electrode, the discharge time was found to be 78.5 s. Based on this discharge time, the specific capacitance was calculated to be 39.4 F g^−1^ (Figure 7a). In Figure 7b–d, it is observed that the charge–discharge time of the fabricated Mn_3_O_4_@NPC/CC electrodes decreased as the current density increased, indicating that a shorter time was required for the charging and discharging processes. The shape of the charge–discharge curve resembles a triangle, which indicates a better Coulombic efficiency [48]. At a current density of 1 A g^−1^, the discharge times of the Mn_3_O_4_@NPC/CC electrodes prepared with low, middle, and high P-dopant amounts were 382.8 s, 381 s, and 515.1 s, respectively (Figure 7e). The curves showing the discharge behavior exhibit four distinct stages. The first stage corresponds to the initial potential drop at the beginning of the discharge curve. The second stage shows a linear change in the curve, indicating electrochemical double-layer capacitance (EDLC) behavior. The third stage is characterized by a pull-off phenomenon in the curve, indicating pseudo-capacitive behavior. Finally, the curve exhibits a drop, indicating redox reactions during the electrochemical reaction [49]. These four stages of discharge behavior indicate a combination of EDLC and pseudo-capacitive behavior within the Mn_3_O_4_@NPC/CC electrodes, thereby contributing to their overall electrochemical performance.

The specific capacitance values of the different electrodes were determined by analyzing the discharge curves obtained from the GCD tests, as depicted in Figure 7f. The Mn_3_O_4_@NPC/CC electrodes, prepared with low, middle, and high amounts of P-dopant, exhibited specific capacitance values of 191.8 F g^−1^, 190.9 F g^−1^, and 256.8 F g^−1^ at a current density of 1 A g^−1^, respectively. The enhanced capacitance can be attributed to the hybridization of Mn_3_O_4_ with the NPC polycrystalline phase, which was facilitated by the introduction of N/P atoms. This doping altered the surface wettability of the Mn_3_O_4_-based electrode, as demonstrated by the contact angle measurements (Figure 2b). Additionally, the increased specific surface area and microporous area of the Mn_3_O_4_-based material contributed to a higher ion storage capacity within the electrode’s pores. The greater the storage capacity, the higher the observed capacitance value.

In order to evaluate the electrochemical performances of the Mn_3_O_4_@NPC/CC electrodes, electrochemical stability and impedance analyses were conducted, as depicted in Figure 8. The capacitance retention of the fabricated electrodes was compared after 5000 charge/discharge cycles at a current density of 4 A g^−1^.

It was observed that the Mn_3_O_4_/CC electrode experienced a significant decrease in its capacitance retention to less than 50% after approximately 127 cycles of charging and discharging. This decline can be attributed to the dissolution or detachment of the Mn_3_O_4_ nanoparticles from the electrode surface or carbon fiber cloth in the electrolyte (Figure 8a). In this study, the CC substrate was immersed in the Mn(OH)_2_ solution without any adhesive material to obtain the Mn_3_O_4_/CC electrode. The Mn_3_O_4_ on the surface of the electrode was easily peeled off during the cyclic stability test. However, after adding carbon materials, the as-fabricated Mn_3_O_4_@NPC/CC electrode significantly improved this situation. The initial capacitance retention rates of the Mn_3_O_4_/CC, Mn_3_O_4_/NPC(L)/CC, Mn_3_O_4_/NPC(M)/CC, and Mn_3_O_4_@NPC(H)/CC electrodes were 42.3%, 91.3%, 89.3%, and 97.3%, respectively. The Mn_3_O_4_@NPC(H)/CC electrode exhibited an excellent cycle stability with a high initial capacitance retention rate. This improved stability can be attributed to the utilization of chitosan as the carbon-based precursor, which imparted stable chemical properties and electrical conductivity to the carbon material. The presence of chitosan helped to prevent the continuous accumulation of Faradaic charges in the composite nanomaterial, thereby maintaining its capacitance value and enhancing the electrochemical stability of the material [50].

As shown in Figure 8b, the Coulombic efficiency of the Mn_3_O_4_@NPC(H)/CC electrode hybrid electrode was excellent, as it was as high as 87.7%, revealing the reversibility of the adsorbed and desorbed electrons on it. Electrochemical impedance spectroscopy (EIS) was performed to investigate the electrochemical reaction kinetics of the Mn_3_O_4_-based/CC electrodes, and the results are shown in Figure 8c. In the Nyquist plot, the impedance curve exhibits a sharp increase and tends toward a vertical line in the low-frequency region, indicating pure capacitive behavior [51]. In the high-frequency region (100 kHz to 10 MHz), the intercept of the Z′ axis represents the equivalent series resistance (ESR), which encompasses the internal resistance of the electrode material, the ionic resistance of the electrolyte, and the contact resistance at the active material/current collector interface. The ESR values for the Mn_3_O_4_/CC, Mn_3_O_4_@NPC(L)/CC, Mn_3_O_4_@NPC(M)/CC, and Mn_3_O_4_@NPC(H)/CC electrodes were 8.89 Ω, 5.48.5 Ω, 4.55 Ω, and 3.86 Ω, respectively.

In the middle-frequency region, a semicircle is observed on the plot, representing the charge transfer resistance (R_ct_) at the electrode/electrolyte interface. The charge transfer resistance (R_ct_) values for the Mn_3_O_4_/CC, Mn_3_O_4_@NPC(L)/CC, and Mn_3_O_4_@NPC(M)/CC electrodes were 6.64 Ω, 0.74 Ω, and 0.59 Ω, respectively. Particularly, it was very hard to find the semicircle on the plot of the Mn_3_O_4_@NPC(H)/CC electrode. These impedance results provide insights into the electrochemical behavior of the electrodes. The lower ESR and R_ct_ values for the Mn_3_O_4_@NPC/CC electrode indicate improved charge-transfer kinetics and a reduced resistance at the electrode/electrolyte interface. This suggests an enhanced electrochemical performance and faster ion transport within the electrode [51].

The absence of a semicircle in the high-to-intermediate-frequency region for the Mn_3_O_4_@C/CC and Mn_3_O_4_@NPC(H)/CC electrodes suggests a rapid reaction between the electrode surface and the electrolyte. This behavior is indicative of improved kinetics and efficient charge transfer processes at the electrode/electrolyte interface. The Mn_3_O_4_@NPC(H)/CC electrode exhibited the lowest ESR and R_ct_ values, indicating a tight connection between the electrode materials [52].

Table 2 shows a comparison of the electrochemical performances of the Mn_3_O_4_-based hybrid electrodes in the literature [20,24,26,53,54]. The GCD test results show that the composite with hybrid Mn_3_O_4_/NPC(H)/CC demonstrated a specific capacitance of 256.8 F g^−1^ at 1 A g^−1^ in 1 M Na_2_SO_4_ electrolyte and an excellent capacitance retention of 97.3% at 5000 charge/discharge cycles. The electrochemical performance of the as-prepared Mn_3_O_4_/NPC(H)/CC electrode was higher than that of previously studied electrode materials.

In this study, the presence of N/P atoms embedded in the carbon framework enhanced the conductive pathways within the Mn_3_O_4_@NPC(H)/CC electrode, thus facilitating electrostatic attraction and providing numerous electroactive sites. These findings are consistent with the observed superior capacitive performance of the Mn_3_O_4_@NPC(H)/CC electrode at high scan rates and current densities. Overall, the tight connection and improved conductive pathways contributed to the enhanced electrochemical behavior of the Mn_3_O_4_@NPC(H)/CC electrode.

## 4. Conclusions

In this study, a chitosan biomass material was utilized as a precursor for a carbon skeleton. Through a hydrothermal process, an Mn_3_O_4_@NPC/CC electrode with exceptional capacitive properties was synthesized from N, P-codoped carbon crystalline hybridized with Mn_3_O_4_ on a carbon fiber cloth (CC) substrate.

The morphology of the electrode material was transformed from a cluster structure to a flower-like structure due to the introduction of N, P-codoped carbon polycrystalline. Additionally, the incorporation of NPC not only reduced the crystallinity of the Mn_3_O_4_, but also optimized the micropore/mesopore ratio, creating a favorable environment for the storage, transmission, and diffusion of electrolyte ions within the Mn_3_O_4_@NPC/CC electrode.

The fabricated Mn_3_O_4_@NPC(H)/CC hybrid exhibited pseudo-capacitive behavior. It maintained a high Coulombic efficiency at low scan rates, and the capacitance significantly increased to 256.8 F g^−1^ at 1 A g^−1^, with a capacitance retention rate of 97.3% over 5000 charge/discharge cycles for the Mn_3_O_4_@NPC/CC electrode with a higher P-dopant content. These results validate the excellent electrochemical properties of N, P-codoped carbon crystalline hybridized with Mn_3_O_4_, highlighting the potential for its application in flexible supercapacitor electrodes.

## Data Availability

Not applicable.

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
