# Peer review of "Boosting the Capacitive Performance of Supercapacitors by Hybridizing N, P-Codoped Carbon Polycrystalline with Mn3O4-Based Flexible Electrodes"

_nanomaterials, 2023, doi:10.3390/nano13142060_

Round 1
Reviewer 1 Report
Research article by Yu-Min Kang et al. as " Boosted Highly Capacitive Performance by N, P-Codoped Carbon Polycrystalline Hybridized with Mn3O4-Based Flexible Electrodes for Supercapacitor" contains a new method to synthesize N, P-doped carbon materials hybridized with Mn3O4 for supercapacitor. After carefully reading, it was found that this article needs major revision because several issues and explanations are still need to be clarified. I recommend it publication in this journal after providing proper improvement in revised version by including suggestion, modification and reply to raised queries which are given below.
1. Why does acetic acid promote the dissolution of chitosan? Why choose phosphoric acid as phosphorus source? Does phosphoric acid affect the dissolution of chitosan? Please give some reasons.
2. The two-electrode system test should be added. And the results of this work should be compared with other researches.
3. Is the mass of Mn3O4 loaded in each sample the same? Will the different mass of Mn3O4 affect the specific capacitance?
4. Why does the specific capacitance of the samples drop so fast?
5. Why does the cyclic stability of Mn3O4/CC drop dramatically to less than 50%?
6. Introduction section is suggested to be improved by adding more recent and relevant articles, such as Journal of Bioresources and Bioproducts 2022, 7 (1), 63-72; Materials & Design 2023, 229, 111904; Colloids and Surfaces A: Physicochemical and Engineering Aspects 2023, 668, 131425.
7. There are some grammatical and punctuation errors in this manuscript. The English language should be improved. Tenses are not consistent from sentence to sentence and there are some grammatical errors.
8. In this work, the specific capacitance of Mn3O4@NPC(H)/CC is the largest. If the phosphoric acid content increases, will the specific capacitance continue to increase? More experiments should be performed.
Moderate editing of English language is required.
Reviewer 2 Report
In the manuscript entitled "Boosted Highly Capacitive Performance by N, P-Codoped Carbon Polycrystalline Hybridized with Mn3O4-Based Flexible Electrodes for Supercapacitor", authors have prepared composite materials containing Mn3O4 supported on chitosan-derived carbon and carbon cloth for energy storage applications.
The manuscript is well written, tidy, and the research design is good. However, the authors should improve a few things in the manuscript to improve it. Please find below my comments and questions.
1) First, the authors should emphasize the novelty of their research
2) In the Introduction part, the third paragraph, which begins with "However, carbon materials..." In my opinion, "However" does not look very appropriate, since the authors in the previous paragraph were talking about manganese.
3) P.3. Section 2.1. The authors should specify the heating rate.
4) P.4. Section 2.2. The authors should mention the material of the cathode as a source of X-ray radiation. And also the type of baseline when fitting spectra should be indicated.
5) The authors should describe the method for determining the contact angle. The photographs do not show the powder. What kind of material was taken?
6) P.6. When analyzing XRD patterns, the authors repeatedly use the concept of crystallinity: "low crystallinity", "good crystallinity". In my opinion, it is not quite appropriate here. The authors need to look at the results from the other side. The change in the peaks of Mn3O4 particles indicates a decrease in their particle size. I think the authors will not have a problem to explain the reason for the decrease of their size and to complete the description of the X-ray analysis.
7) P.7. L.26. Is there any mix-up between the terms "lattice period" and "lattice spacing"? In the microphotographs, the authors measure the lattice spacing.
8) P.8. Since the authors assign an important role to the carbon component of the resulting material, they should also give XPS region C1s.
9) P.8. The fitting of the O1s region raises questions, since all of the indicated bonds are exclusively with manganese. The authors assume that the carbon support has no oxygen groups?
10) P.8. The attribution of the 529.6 eV peak as Mn-O-C also raises questions. The authors should support this statement with a reference to the relevant publications.
11) P.8. L 30. Not for the first time, the authors claim that chitosan gives N for the synthesis of Mn3O4. What role does nitrogen play in the synthesis of Mn3O4?
12) The authors should provide an elemental analysis of the obtained materials. Maybe these data will free the authors from the need to give a very noisy (due to low nitrogen content) spectrum of the N1s region. And the authors should give the content of manganese in the final materials.
13) P.9. L.33. The authors' claim that BET analysis allowed to determine the synthesis of the Mn2O4@NPC material raises serious questions. Nitrogen adsorption/desorption cannot say anything about this.
14) As for the change in surface area when H3PO4 is introduced. Don't the authors suggest that this acid can act as an activating agent, thereby increasing the porosity?
15) What is the reason for dividing the electrochemical studies into two sections: "3.2. Electrochemical Properties of Mn3O4@NPC/CC Electrodes" and "3.3. Electrochemical Performances of Mn3O4@NPC/CC Electrodes"?
16) P.11. L. 41. Authors should correct typos and bring the names of samples to the same format.
17) P.13. In the Conclusion section, the authors again mention crystallinity. Perhaps this interpretation of the results should be reconsidered.
18) Just wondering if I'm right in feeling that the authors used ChatGPT to polish the text?
Round 2
Reviewer 1 Report
The manuscript could be accepted now.